# In Vitro and In Vivo Evaluation of Human Adenovirus Type 49 as a Vector for Therapeutic Applications

**DOI:** 10.3390/v13081483

**Published:** 2021-07-28

**Authors:** Emily A. Bates, John R. Counsell, Sophie Alizert, Alexander T. Baker, Natalie Suff, Ashley Boyle, Angela C. Bradshaw, Simon N. Waddington, Stuart A. Nicklin, Andrew H. Baker, Alan L. Parker

**Affiliations:** 1Division of Cancer and Genetics, School of Medicine, Cardiff University, Cardiff CF14 4XN, UK; BatesE@Cardiff.ac.uk (E.A.B.); Baker.Alexander@mayo.edu (A.T.B.); 2Genetics and Genomic Medicine Research and Teaching Department, UCL Great Ormond Street Institute of Child Health, London WC1N 1EH, UK; j.counsell@ucl.ac.uk; 3NIHR Great Ormond Street Hospital Biomedical Research Centre, 30 Guilford Street, London WC1N 1EH, UK; 4Institute of Cardiovascular and Medical Sciences, BHF Glasgow Cardiovascular Research Centre, University of Glasgow, Glasgow G12 8TA, UK; sophie.alizert84@gmail.com (S.A.); Angela.Bradshaw@glasgow.ac.uk (A.C.B.); Stuart.Nicklin@Glasgow.ac.uk (S.A.N.); 5Center for Individualized Medicine, Mayo Clinic, Scottsdale, AZ 85259, USA; 6Department of Women and Children’s Health, King’s College London, St Thomas’ Hospital, Westminster Bridge Road, London SE1 7EH, UK; natalie.suff@kcl.ac.uk; 7Gene Transfer Technology Group, EGA Institute for Women’s Health, University College London, 86-96 Chenies Mews, London WC1E 6BT, UK; ashley.boyle@ucl.ac.uk (A.B.); s.waddington@ucl.ac.uk (S.N.W.); 8MRC Antiviral Gene Therapy Research Unit, Faculty of Health Sciences, University of the Witswatersrand, Johannesburg 2193, South Africa; 9Queen’s Medical Research Institute, University of Edinburgh, 47 Little France Crescent, Edinburgh EH16 4TJ, UK

**Keywords:** adenovirus, viral vector, gene therapy, vaccines

## Abstract

The human adenovirus phylogenetic tree is split across seven species (A–G). Species D adenoviruses offer potential advantages for gene therapy applications, with low rates of pre-existing immunity detected across screened populations. However, many aspects of the basic virology of species D—such as their cellular tropism, receptor usage, and in vivo biodistribution profile—remain unknown. Here, we have characterized human adenovirus type 49 (HAdV-D49)—a relatively understudied species D member. We report that HAdV-D49 does not appear to use a single pathway to gain cell entry, but appears able to interact with various surface molecules for entry. As such, HAdV-D49 can transduce a broad range of cell types in vitro, with variable engagement of blood coagulation FX. Interestingly, when comparing in vivo biodistribution to adenovirus type 5, HAdV-D49 vectors show reduced liver targeting, whilst maintaining transduction of lung and spleen. Overall, this presents HAdV-D49 as a robust viral vector platform for ex vivo manipulation of human cells, and for in vivo applications where the therapeutic goal is to target the lung or gain access to immune cells in the spleen, whilst avoiding liver interactions, such as intravascular vaccine applications.

## 1. Introduction

Human adenoviruses are non-enveloped, icosahedral viruses, divided across seven species (A–G) [1]. They have emerged as popular gene therapy vectors for therapeutic purposes, where they can be grown to high titers and efficiently transduce a range of cell types in vivo and in vitro [2]. Their diverse genetic background enables flexibility when selecting capsid serotypes with unique cell targeting profiles and host interaction characteristics, depending on the therapeutic goal of gene transfer.

Adenoviruses have been used clinically for gene supplementation, vaccination, and as oncolytic virotherapy [3]. To date, there have been over 100 adenovirus types described (http://hadvwg.gmu.edu/, accessed on 10 June 2021). When selecting an adenoviral vector for in vivo gene therapy applications, it is advantageous to select a viral platform with low levels of seroprevalence in the population. Such pre-existing immunity would otherwise hamper the efficacy of adenovirus vectors, due to rapid neutralization by the reticuloendothelial system. This is particularly relevant to species C adenoviruses, such as human adenovirus type 5 (HAdV-C5), for which neutralizing antibodies have been detected at rates of 30–90% in human populations [4,5], which result in the rapid sequestration and elimination of therapeutics based on HAdV-C5 [6]. For intravenous applications, HAdV-C5 is also hampered by interactions with host proteins, which result in efficient elimination by the liver and spleen (reviewed in [7,8]). Critical amongst such interactions is the high-affinity, Ca^2+^-dependent interaction between the major adenoviral capsid protein—the hexon—and circulating blood clotting factor X (FX), which results in efficient, heparan sulphate proteoglycan (HSPG)-dependent transduction of liver hepatocytes [9,10,11]. Circumventing such interactions in the blood has required extensive genetic engineering approaches to develop heavily modified HAdV-C5-based vectors better suited to targeted intravenous approaches [12,13]. An alternative approach is to develop viral platforms with alternative receptor usage [14] and more limited interactions with blood clotting factors, such as those derived from species D [10].

Several groups have attempted to circumvent this restriction by selecting adenovirus capsids with naturally low seroprevalence rates, such as the chimpanzee adenovirus platform developed by The Jenner Institute for vaccination against SARS-CoV-2 [15,16]. However, it has been reported that some populations do harbour pre-existing immunity to chimpanzee adenoviruses, as observed in a Chinese cohort [17]. Further attempts to mitigate pre-existing immunity against adenoviruses has focused on those derived from species B or D, due to their comparative rarity in human populations [4,18,19]. This includes the species D member HAdV-D26, which is the basis of the Janssen Pharmaceuticals vaccine against SARS-CoV-2 [20], and the species B variant enadenotucirev (ColoAd1), which was developed by directed evolution of a panel of different adenovirus strains [21].

Adenovirus type 49 (HAdV-D49) is a species D member, which has also been evaluated for potential for gene therapy applications in vitro and ex vivo, owing to its particularly low seroprevalence rates. A study of a Scottish cohort failed to detect any neutralizing antibodies to HAdV-D49 at all [5], although low levels of 1–2% were detected in further screening studies in Europe [22,23]. Previous studies have highlighted HAdV-D49 as a potential vaccine vector [24], and as an effective agent for ex vivo cardiovascular gene delivery due to its efficiency in rapidly transducing endothelial and vascular smooth muscle cells [5]. Previous studies into the basic virology of HAdV-D49 suggest that it may engage CD46 as a cellular receptor [24], although its exact entry mechanism remains unclear, and a recent study indicated that the highly charged fibre knob protein may provide a novel mechanism for cell entry [25]. Overall, there remain limited investigations into the basic biology of HAdV-D49 and its human cell transduction characteristics, despite its described advantages.

Here, we have evaluated the transduction characteristics and tropism of HAdV-D49 for in vitro and in vivo manipulation of cells, to better evaluate its potential use and therapeutic exploitation.

## 2. Materials and Methods

### 2.1. Cells and Tissues

HEK 293T cells (human embryonic kidney: ATCC CRL-1573) were used for viral production, cultured in Dulbecco’s modified Eagle’s medium (DMEM; Invitrogen, Grand Island, NY, USA) supplemented with 2 mM l-glutamine (Invitrogen) and 10% foetal bovine serum (FBS; PAA Laboratories, Cölbe, Germany). HepG2 (hepatocellular carcinoma: ATCC HB-8065), A549 (human lung carcinoma: ATCC CCL-185), SKOV3 (human ovarian carcinoma: ATCC HTB-77), HeLa (cervical adenocarcinoma cells: ATCC CCL-2), HT-29 (colon adenocarcinoma cells: ATCC HTB-38) and MDA435 (melanocyte: ATCC HTB-129) cells were cultured in DMEM, minimal essential medium (MEM), or RPMI-1640 medium (Invitrogen with 2 mM L-glutamine, 10% FCS, and 1 mM sodium pyruvate (Sigma-Aldrich, St. Louis, MO, USA)). Cells were maintained at 37 °C and 5% CO_2_.

### 2.2. Adenovirus Culture

HAdV-C5 and HAdV-D49 are replication-incompetent E1/E3-deleted vectors constructed as described previously [14]. HAdV-C5/49 fibre (49F) and fibre knob (49K) pseudotypes and HVR mutant versions were generated using previously described recombineering methods [23]. Viruses were propagated in HEK 293T, E1-complementing cell lines and purified using CsCl gradients. Viral recovery was quantified by micro-BCA assay (Thermo Fisher, Loughborough, UK), assuming that 1 g protein = 4 × 10^9^ viral particles (vp), and confirmed by NanoSight measurement (NanoSight, Malvern, UK). Infectious units (pfu) were quantified by end-point dilution plaque assay.

### 2.3. Luciferase Transduction Assay

Assay was performed using a commercially available luciferase assay kit (Promega). Cells were seeded into 96-well cell plates at a density of 2 × 10^4^ cells/well in 200 μL of cell culture media and left to adhere overnight. Viruses were administered in relevant media and incubated for 3 h, before replenishing with complete media and culturing for a further 45 h. Cells were then lysed, with lysates assayed for luciferase activity according to the kit manufacturer’s instructions. Luciferase activity was measured in relative light units (RLUs) using a plate reader (CLARIOstar, BMG Labtech, Aylesbury, UK). Total protein concentration was determined in the lysate using the Pierce BCA protein assay kit (Thermo Fisher, Loughborough, UK) according to the manufacturer’s protocol. Transduction efficiency was expressed as RLU/mg of lysate protein.

### 2.4. Heparinase and Neuraminidase Transduction Assay

Cells were seeded at a density of 5 × 10^4^ cells/well in 96-well plates. Neuraminidase (from Vibrio Cholera, Merck, Darmstadt, Germany) was added at a concentration of 50 mU/mL, whilst heparinase III (from Flavobacterium heparinum, Merck, Darmstadt, Germany) was added at a concentration of 1 U/mL, diluted in serum-free media. Adenoviruses carrying a luciferase transgene were added in serum-free media and incubated on ice for 1 h. Cells were then washed with cold PBS and replenished with complete media, before returning to incubation at 37 °C for 48 h. Cells were analysed for luciferase activity as described above.

### 2.5. Hemagglutination Assay

Erythrocytes were extracted from blood cells derived from a human donor, who gave informed consent. Then, 50 µL of 1% erythrocyte suspension was layered in each well of a 96-well plate, before adding 50 µL of relevant virus (1 × 10^9^ VP) and gently mixing into the erythrocyte suspension. Haemagglutination was assessed visually.

### 2.6. Generation of Recombinant Fibre Knob Proteins

Recombinant fibre knob proteins were produced as described previously [14]. pQE-30 vectors, containing the coding sequence spanning 13 amino acids upstream of the TLW motif to the stop codon, were transformed into SG13009-competent cells harbouring the pREP-4 plasmid. Then, 1 L of bacterial cells was grown to OD0.6, and protein expression was induced with a final concentration of 0.5 mM IPTG. Cells were pelleted by centrifugation and resuspended in 50 mL lysis buffer (50 mM Tris, pH 8.0, 300 mM NaCl, 1% (*v*/*v*) NP40, 1 mg/mL Lysozyme, 1 mM β-mercaptoethanol). Cell lysate was then loaded to a HisTrap FF crude column and eluted with imidazole. Fractions determined to contain proteins of interest were then concentrated to <1 mL total volume and purified by size exclusion chromatography using a Superdex 200 10/300 GL increase column. Validation of recombinant knob trimerisation was performed with 5 µg and 10 µg using Coomassie staining. 

### 2.7. Blocking of Virus Infection with Recombinant Fibre Knob Protein

Cells were seeded in 96-well plates at a density of 2 × 10^4^ cells/well. The relevant adenovirus fibre knob was added to wells in 200 μL of cold PBS and incubated at 4 °C for 1 h. Media were then removed and adenovirus vectors containing luciferase payloads were introduced at the required dose in culture media before incubating at 4 °C for 1 h. Media were then removed, replaced with complete media, and cells incubated at 37 °C for 48 h. Transduction efficiency was measured by luciferase assay, as described above.

### 2.8. Animals

For in vivo studies, animal procedures were performed in strict accordance with UK Home Office guidelines. These studies were approved by the University of Glasgow Animal Procedures and Ethics Committee and performed under UK Home Office license PPL 60/4429, or by the ethical review committee of University College London under UK Home Office License PPL 70/6014. All efforts were made to minimize suffering. 

For intravenous studies, outbred CD1 mice (Charles River Laboratories International, USA) were injected intravenously with the stated doses of adenoviral vectors and blinded during the course of in vivo investigations. Foetal intracranial injection was performed as previously described [26,27]. Briefly, pregnant mice carrying pups at 16 days of gestation were anaesthetized using isoflurane inhalation anaesthesia, and a midline laparotomy was performed to expose the uterus. Five microliters of vector was administered to each foetus via a transuterine injection targeting the anterior horn of the lateral ventricle of the left hemisphere of the brain. The laparotomy was sutured and mice were provided topical and systemic analgesia and allowed to recover in a warm chamber. 

### 2.9. In Vivo Bioluminescent Imaging

Images and bioluminescence data were gathered as described previously [28]. Firefly D-luciferin (150 mg/kg) was administered to mice by intraperitoneal injection 5-min before imaging with a cooled charge-coupled device (CCD) camera (IVIS Lumina II, PerkinElmer). Detection of bioluminescence in visceral organs was performed using the auto region of interest (ROI) quantification function in Living Image 4.4 (PerkinElmer). Signal intensities were expressed as photons per second per centimeter^2^ per steradian.

### 2.10. Luminex Quantification of Cytokine and Chemokine Levels

HAdV-C5 or HAdV-D49 (1 × 10^11^ vp) vectors encoding luciferase were administered by intravenous injection into outbred CD1 mice. Then, 50 μL of blood was extracted by venesection and collected in capillary tubes 6 h following virus administration. Serum was prepared by allowing blood to coagulate at room temperature for 30 min, then centrifuging at 10,000× *g* rpm for 15 min. Cytokine and chemokine analysis of sera was performed using a mouse cytokine 20-plex Luminex panel according to the manufacturer’s instructions (Invitrogen, Paisley, UK), quantifying levels of basic fibroblast growth factor (bFGF), granulocyte macrophage colony-stimulating factor (GM-CSF), interferon-gamma (IFN-γ), interleukin (IL)-1α, IL-1β, IL-2, IL-4, IL-5, IL-6, IL-10, IL-12(p40/p70), IL-13, IL-17, IFN-induced protein (IP10), keratinocyte-derived cytokine (KC), monocyte chemoattractant protein (MCP-1), monokine induced by gamma interferon (MIG), macrophage inflammatory protein-1alpha (MIP-1α), tumour necrosis factor-alpha (TNF-α), and vascular endothelial growth factor (VEGF). Data were analysed using Bio-Plex manager software with 5PL curve fitting. 

### 2.11. Calculation of Vector Genome Copies in Mouse Organs

Genomic DNA was extracted from the organ of interest following tissue harvest after termination of in vivo experiments. DNA was extracted from tissue using the Qiagen DNeasy blood and tissue kit. Vector genome copies were quantified by qPCR, targeted to the adenoviral vector genome, and expressed per 50 ng of extracted DNA.

### 2.12. Statistical Analysis

Data presented are derived from a minimum of two experimental replicates per group, unless otherwise stated. Transduction graphs are displayed with a log scale, and for these experiments data were log transformed before analysis. Statistical significance was calculated using ANOVA or Student’s *t*-test; *p* < 0.05 was considered statistically significant.

## 3. Results

Our experiments focused on the in vitro and in vivo transduction characteristics of HAdV-D49 vectors, with relevance to gene therapy applications.

### 3.1. HAdV-D49 Transduces a Broad Range of Cell Types In Vitro

We initially profiled the transduction efficiency of HAdV-D49 in a range of cell types to gauge its potential use for cellular gene transfer in vitro. To evaluate the efficiency of transduction, we performed assays at two doses of HAdV—a standard dose of 1000 vp/cell (Figure 1A), and a higher dose of 10,000 vp/cell (Figure 1B). Expression of luciferase transgene highlighted the efficient transduction of several immortalized cell types mediated by HAdV-D49, including alveolar basal endothelial cells (A549 cells), colon adenocarcinoma cells (HT29), cervical adenocarcinoma cells (Hela), melanocytes (MDA435), and ovarian carcinoma cells (SKOV3). In the majority of cases, HAdV-D49 demonstrated a significantly increased transduction efficiency compared to HAdV-C5.

In light of the above findings, we investigated whether FX binding could further enhance HAdV-D49-mediated transduction. A previous study indicated that HAdV-D49 forms an unstable complex with FX [10] and, therefore, it remained unclear how this unstable complex might impact on cellular transduction efficiency. Cells were transduced with either HAdV-D49 or HAdV-C5 vectors expressing luciferase in serum-free media, in the presence or absence of physiological concentrations of FX (10 μg/mL), with expression determined by bioluminescent signal from the luciferase transgene (Figure 1C). HAdV-D49 transduction was only significantly increased by the presence of FX in MDA 435 cells, possibly reflecting high expression of heparan sulphate proteoglycan (HSPGs)—known to mediate FX-mediated viral cell entry—in MDA-435 cells. FX consistently enhanced the transduction of HAdV-C5, with a significant increase for three cell lines (A549, MDA 435, and SKOV3). This indicates that FX plays a variable role in HAdV-D49 infection; however, the presence of FX is not a requirement for cell entry, and nor does its presence enhance transduction mediated by HAdV-D49 in all cell lines tested.

### 3.2. In Vivo Biodistribution Profiling Shows That FX Interaction Does Not Confer Increased Liver Targeting

We next investigated the in vivo biodistribution of HAdV-D49 vectors in mice, to determine which visceral organs were targeted by systemic delivery. We delivered HAdV-D49 vectors carrying a luciferase transgene to mice via intravenous injection, and quantified vector genome copy numbers and luciferase expression in a range of target organs 48 h later, comparing their distribution to that of HAdV-C5 (Figure 2). As expected, HAdV-C5 vectors showed strong liver tropism (Figure 2A). Despite a possible interaction with FX, HAdV-D49 did not appear to transduce the liver (Figure 2B), but instead demonstrated increased uptake in the spleen (Figure 2C). Quantification of vector genomes recovered from the lung, liver, spleen, heart, kidney, and pancreas support these findings (Figure 2D). HAdV-D49 shows the highest uptake in the lung and spleen, with significantly lower uptake in the liver compared to HAdV-C5. We then considered the effect of significantly higher levels of HAdV-D49 in the spleen on cytokine production (Figure 2E and Appendix A). Evaluation of cytokine profiles from mouse plasma shows that HAdV-D49 treatment significantly increases a panel of inflammatory cytokines, including FGF, MCP1, MIG, IFN-gamma, IL1-beta, IL-2, IL-5, IL-6, IL-12, and IP-10. Increased cytokine activation coupled with reduced liver targeting suggests that HAdV-D49 possesses advantages over HAdV-C5 for use in gene therapy or vaccine applications. 

### 3.3. HAdV-D49 Fibre Alone Does Not Mediate Liver Transduction In Vivo

To further dissect the mechanistic basis of the decreased hepatic transduction and increased splenic transduction observed following intravascular administration of HAdV-D49 compared to HAdV-C5, we utilised a panel of HAdV-C5/49-pseudotyped vectors (Figure 3). To assess whether this altered distribution results from the interaction between the HAdV-D49 primary receptors and the fibre knob protein, we developed HAdV-C5 vectors pseudotyped with either the fibre knob protein alone from HAdV-D49 (HAdV-C5/K49), or with the whole fibre (HAdV-C5/F49). Since hepatic transduction of HAdV-C5 is known to be mediated by the high-affinity interaction between the HAdV-C5 hexon protein and FX, we also generated mutants with substitutions of critical FX binding residues in HVR7 to abrogate FX binding and, thus, limit liver transduction. As expected, HAdV-C5 vectors with a WT hexon showed higher transduction of the liver compared to those with ablation of the FX binding (HVR mutant). Interestingly, when the whole HAdV-C5 fibre was replaced by HAdV-D49 whole fibre, no uptake in the liver was observed for either the WT hexon or the HVR mutant. However, liver transduction was restored when only the HAdV-D49 fibre knob was swapped. No liver transduction was observed in HAdV-C5/K49 when HVR was mutated; this indicates that the fibre alone does not mediate splenic transduction, and HAdV-D49 most likely requires additional components, beyond its fibre, to define its tropism in vivo. In addition, this data suggests that the HAdV-D49 fibre knob does not significantly alter HAdV-C5 biodistribution when administered intravenously, and is able to mediate transduction in the liver. 

### 3.4. HAdV-D49 Show Similar Levels of Intracranial Transduction to HAdV-C5

In order to evaluate the potential of HAdV-D49 as a platform for neurological gene transfer applications, MF1 mice were injected intracranially in utero with 1 × 10^9^ vp of either HAdV-C5- or HAdV-D49-expressing luciferase. At three months of age, mice were IVIS imaged for luciferase activity (Figure 4). Mice treated with HAdV-D49.Luc mediated readily detectable levels of transduction in the brain, although the levels achieved were lower than those treated with HAdV-C5. 

### 3.5. HAdV-D49 Infection of Target Cells Does Not Appear to Be Attributed to a Single Cell Surface Marker

We have demonstrated that HAdV-D49 exhibits a broad tropism both in vitro and in vivo; we therefore aimed to further our understanding of the mechanism underpinning HAdV-D49 receptor usage. We initially investigated the extent to which HAdV-D49 can engage CAR for transduction of hepatocytes (HepG2). We first demonstrated that we were able to produce HAdV-C5 and HAdV-D49 recombinant knob proteins in a trimeric state (Figure 5A). For these studies, CAR binding was blocked using recombinant HAdV-C5 knob protein prior to transduction (Figure 5B). As expected, HAdV-C5 knob treatment clearly blocked transduction mediated by HAdV-C5; however, the same treatment had no effect on HAdV-D49 transduction. In addition, recombinant HAdV-D49 knob was also used to block cellular receptors. Interestingly, recombinant HAdV-D49 knob was able to inhibit transduction mediated by HAdV-C5, indicating that HAdV-D49 knob protein can bind and block CAR—albeit less efficiently that recombinant HAdV-C5. Interestingly, recombinant HAdV-D49 knob protein was not able to inhibit HAdV-D49-mediated cell infectivity, except poorly at the highest concentration. This indicates that HAdV-D49 is not solely dependent on binding of its fibre knob protein to cells for transduction of cells, and that other capsid proteins may be critical.

We observed that HAdV-D49 haemagglutinates human serum at high concentrations (10^9^ vp/well), indicating that it may bind to CAR or sialic acid, like other species D adenoviruses [29] (Figure 5C). We investigated this potential targeting of sialic acid as well as heparan sulphate proteoglycans (HSPGs), given that these have been identified as interacting partners for adenovirus transduction [25,29,30,31,32,33]. To probe this, hepatocytes (HepG2 cells) were transduced following heparinase and neuraminidase treatment to cleave HSPGs and sialic acid, respectively. Whilst the removal of HSPGs by heparinase demonstrated a small but significant reduction in HAdV-D49-mediated transduction (Figure 5D), no effect was observed following treatment with neuraminidase, indicating that sialic acid is not essential for transduction (Figure 5E). 

Finally, we investigated CD46 interaction, as it has previously been described as having a potential involvement in HAdV-D49 infection [15,23]. CD46 was blocked on A549, HeLa, and SKOV3 cells using a specific monoclonal antibody (MEM-258) prior to transduction with HAdV-C5, HAdV-D49, and HAdV-B35 (Figure 5F). HAdV-D49 infection was reduced significantly in all three cell lines when CD46 was blocked. HAdV-B35 saw significant decreases in transduction in all three cell lines, whereas blocking CD46 had a minimal effect on HAdV-C5 infection. These data suggest that CD46 is a possible receptor for HAdV-D49, consistent with the previous data; however, it is not likely to be the sole receptor, as HAdV-D49 remains able to readily transduce cells even when CD46 is blocked.

## 4. Discussion

Adenoviruses have been widely used in gene therapy for a variety of applications, owing to their diversity and efficiency in delivering therapeutic DNA cassettes to the nuclei of cells in vitro and in vivo. Common applications of adenoviruses include oncolytic virotherapy [1], ex vivo cell manipulation [5], and vaccine delivery [16,20,34,35], whilst recently they have been used pre-clinically to mediate in utero gene editing of disease-causing mutations in foetal mice [36]. Clinical translation of adenovirus therapies could be accelerated by greater understanding of their biology and interactions with human proteins. Additionally, many of the current adenovirus platforms are restricted by the prevalence of neutralizing antibodies against adenoviral capsid proteins, which restrict their delivery to the desired cell types in vivo.

Here, we have shown that HAdV-D49 vectors can be used to deliver genes to a wide range of target cells, both in vitro and in vivo. We identified that HAdV-D49 showed enhanced transduction of hepatocytes in vitro in the presence of human serum, which was linked to a potential interaction with human FX. Given previous reports of HAdV-C5 utilizing FX as a partner for enhanced interaction with its CAR receptor [10,37], we expected a similar concept for HAdV-D49, although this was not an absolute requirement. We observed that FX significantly increased HAdV-D49 transduction in MDA-435 cells alone; we consider whether this can be attributed to varying levels of HSPGs on the cells’ surfaces. However, as there was no significant increase observed in other cell lines tested, we conclude that FX is not essential in mediating HAdV-D49 transduction. Furthermore, the fibre knob domain of HAdV-D49 did not block HAdV-D49 entry into hepatocytes, suggesting that further interacting partners may be involved. This reaffirms the apparent complexity of HAdV-D49 tropism and interactions with human cells during transduction. 

Evaluation of HAdV-D49 biodistribution was efficient in mediating gene transfer, predominantly to the spleen, in vivo, with surprisingly low levels of vector genomes detected in the liver of treated mice considering the potential for FX engagement. The high uptake of HAdV-D49 by the spleen was correlated with an increase in inflammatory cytokine release. Overall, this indicates that HAdV-D49 is a versatile vector with potential usage for a range of in vivo and ex vivo gene therapy and vaccine applications.

We extended our studies deeper into the basic biology of HAdV-D49 vectors, by probing the potential mechanism for its transduction of target cells. Previous studies have suggested that HAdV-D49 utilizes CD46 as its primary cellular receptor [24]. Our data suggest that, although CD46 is interacted with, it is not the exclusive receptor. A recent study describes a number of species D serotypes that are able to engage CD46 directly via the hexon [30]. Although this has not yet been described for HAdV-D49, it should be considered as a mechanism for CD46 interaction. Furthermore, we did not see evidence of a single cell adhesion molecule being absolutely required for HAdV-D49 transduction, instead identifying multiple interacting partners that may be utilized for cell entry. Future studies to unpick the complex tropism of HAdV-D49 will require combinations of inhibitors to fully delineate whether HAdV-D49 uses multiple previously described receptors and/or a completely novel means of cell entry. Evidence of potential interactions with CAR, CD46, HSPGs, and FX were all evident in our data, but no single factor was deemed to be necessary and sufficient for transduction. The broad transduction profile of HAdV-D49 is therefore potentially explained by its apparent promiscuous interaction profile, maintaining efficient transduction across a range of cell types by establishing a repertoire of interaction mechanisms, for greater flexibility over other adenovirus types.

A particularly intriguing aspect of our study was the paradoxical cell tropism of HAdV-D49 in vitro versus in vivo, where it showed efficient transduction of many cell types in vitro, including liver cells, whereas in vivo its biodistribution was largely restricted to lung and spleen, with minimal transduction of liver compared to HAdV-C5. Further studies will be required to identify which specific subsets of cells within the spleen are efficiently transduced by HAdV-D49, but it is likely that increased uptake by immune cells within the spleen correlates with the enhanced innate immune responses we noted following intravascular delivery of HAdV-D49 compared to HAdV-C5. Given that all in vitro studies were conducted in human cells in the presence of human serum or human FX protein, it is reasonable to conclude that efficient HAdV-D49 transduction shows species dependency in its interactions with cell receptors and co-receptors. In order to maximize HAdV-D49 utilization in in vivo gene therapy applications, further work should be aimed at interrogating the species requirements of its cellular interactions, in order to identify the crucial determinants for entry. Indeed, in the context of ex vivo gene therapy, HAdV-D49 clearly shows promise as a versatile and robust gene transfer agent, with exciting potential in pre-clinical and clinical applications.

## Figures and Tables

**Figure 1 viruses-13-01483-f001:**
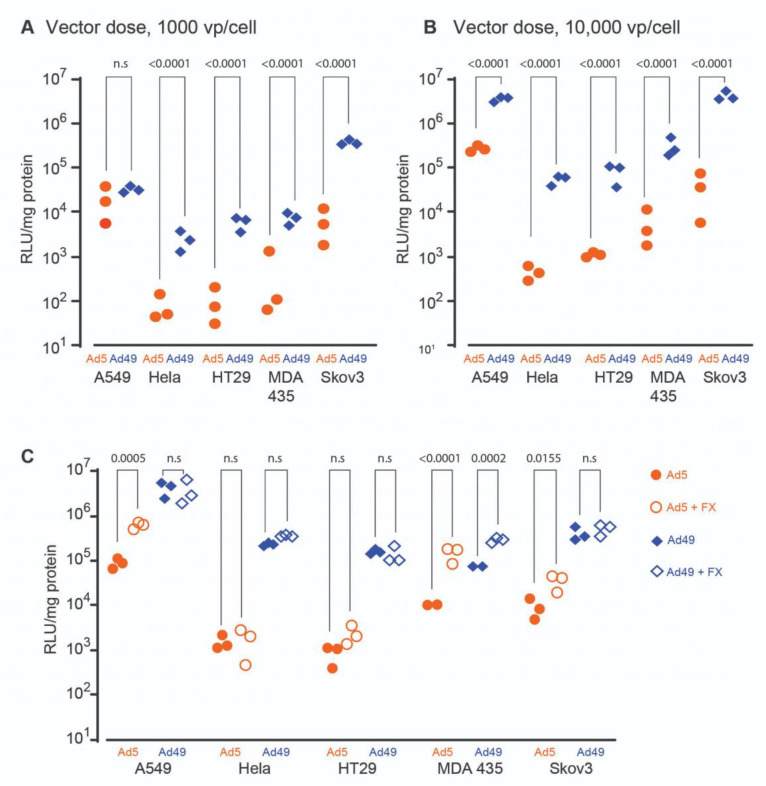
Profiling of HAdV-D49 transduction in cancer cell lines in vitro. Transduction of HAdV-C5 (shown as Ad5) and HAdV-D49 (shown as Ad49) in A549, HeLa, HT20, MDA 435, and SKOV3 cell lines at (**A**) 1000 vp/cell and (**B**) 10,000 vp/cell. Luciferase expression was measured 48 h post-infection. (**C**) Transduction of the same cell lines using 5000 vp/cell, in either the presence or absence of recombinant FX. n.s: not statistically significant, *p* > 0.05.

**Figure 2 viruses-13-01483-f002:**
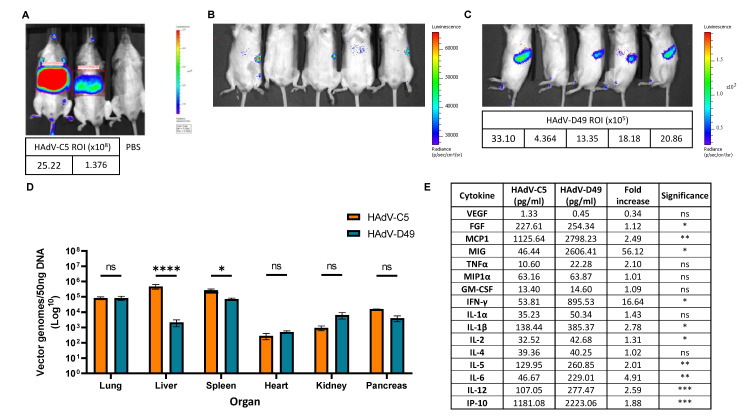
In vivo biodistribution analysis of HAdV-D49 vectors compared to HAdV-C5 vectors. IVIS imaging system used for in vivo quantification of (**A**) HAdV-C5 (ROI ×10^8^) liver transduction with PBS control; (**B**) IVIS images of HAdV-D49 (shown as Ad49) transduction with mice imaged on their backs, and (**C**) IVIS images of HAdV-D49 (Ad49) transduction when mice are imaged on their sides. Uptake in the spleen was quantified using ROI (×10^5^). (**D**) Quantification of biodistribution using a dose of 1 × 10^11^ virus particles. qPCR to evaluate vector genomes in the lung, liver, spleen, heart, kidney, and pancreas. (**E**) Cytokine concentrations, gauged by Luminex 20-plex cytokine analysis in mouse plasma after HAdV-C5 (Ad5) or HAdV-D49 (Ad49) administration (shown as pg/mL), and demonstrated as fold increase of HAdV-D49 in comparison to HAdV-C5. Individual plots are shown in Appendix A. ns, not statistically significant, *p* > 0.05; *, *p* < 0.05; ** *p* < 0.01; ***, *p* < 0.001; ****, *p* < 0.0001.

**Figure 3 viruses-13-01483-f003:**
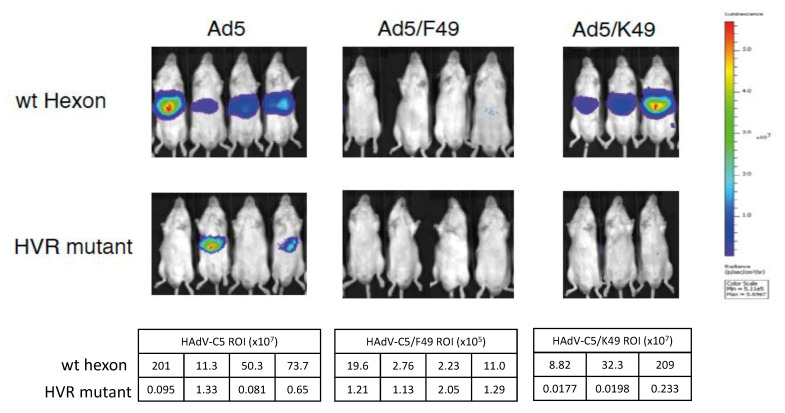
Evaluation of the HAdV-D49 fibre and fibre knob in vivo. Transduction of HAdV-C5 and HAdV-C5/49 fibre (F49) and fibre knob (49K) pseudotypes in vivo, with and without the hexon mutation (HVR mutant), to reduce liver uptake with ROI indicated in tables.

**Figure 4 viruses-13-01483-f004:**
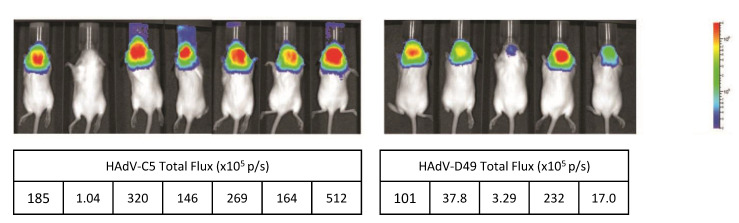
In vivo bioluminescence imaging of luciferase expression mediated by HAdV-C5 (Ad5) vectors compared to HAdV-D49 (Ad49) vectors after foetal intracranial injection. IVIS imaging system was performed at three months of age. Quantification of bioluminescence is shown in table (total flux ×10^5^ p/s).

**Figure 5 viruses-13-01483-f005:**
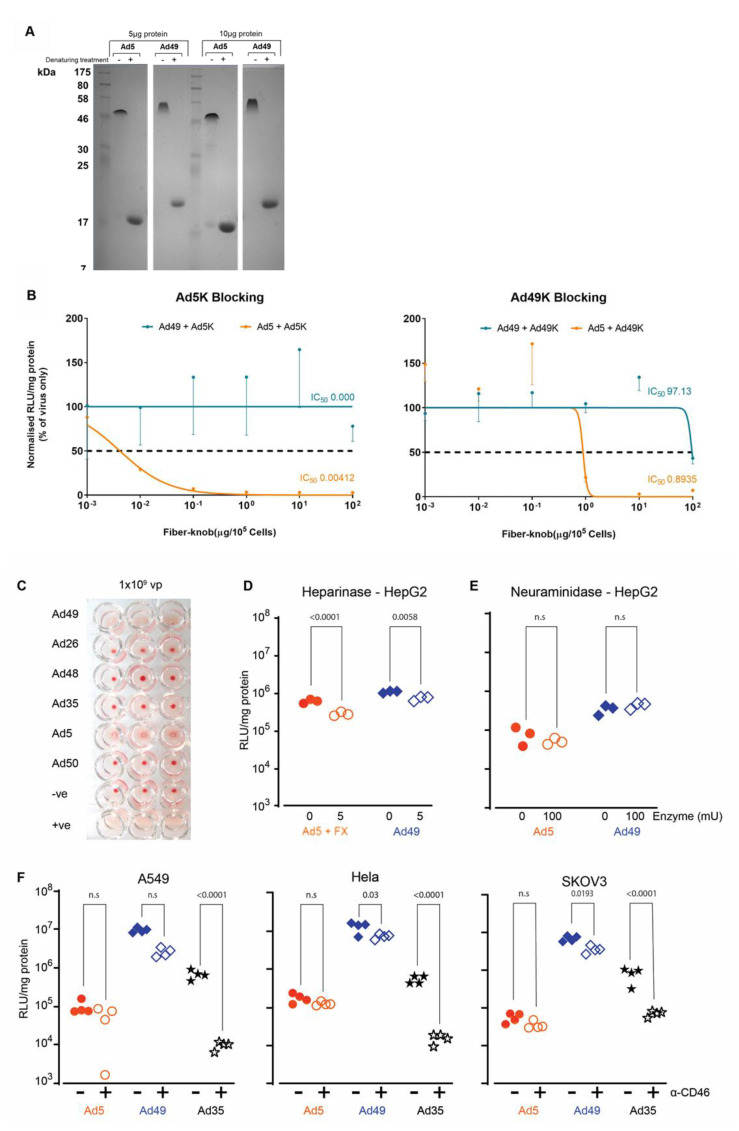
HAdV-D49 can interact with multiple receptors to transduce target cells. (**A**) Recombinant HAdV-C5K and HAdV-D49K with or without treatment using a denaturing agent were stained with Coomassie. (**B**) Blocking of HAdV-C5 (Ad5) and HAdV-D49 (Ad49) transduction in hepatocytes (HepG2) with dilution of HAdV-C5 or HAdV-D49 fibre knob (100–0.001 µg/10^5^ cells). (**C**) Haemagglutination profiling of several adenovirus serotypes, including HAdV-C5 and HAdV-D49. Negative (-ve, PBS) and positive (+ve, PBS containing 0.2% Tween-20) were also used as relevant controls; three wells are shown reflecting technical replicates. (**D**) HAdV-C5 + FX and HAdV-D49 transduction in HepG2 cells after treatment with heparinase. (**E**) HAdV-C5 (shown as Ad5) and HAdV-D49 transduction in HepG2 cells after treatment with neuraminidase. (**F**) Transduction of HAdV-C5, HAdV-D49, and HAdV-B35 (Ad35) in A549, HeLa, and SKOV3 cells following blocking with 258-MEM, anti-CD46 antibody. n.s; not statistically significant, *p* > 0.05.

## Data Availability

The datasets generated during and/or analysed during the current study are available from the corresponding authors on reasonable request.

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
