# Peer review of "In Vitro and In Vivo Evaluation of Human Adenovirus Type 49 as a Vector for Therapeutic Applications"

_viruses, 2021, doi:10.3390/v13081483_

Round 1
Reviewer 1 Report
Adenovirus (Ads) have a well-defined and low pathogenic profile paired with technological accessibility. This makes adenoviruses powerful therapeutic tools for vector-based strategies. In contrast adenoviruses infections are widespread and a large part of the population has pre-existing immunity (antibodies) against the most common serotypes, notably species C adenoviruses. This can conflict with adenovirus vector applications, a problematic that will be further increased through the use of adenovirus as anti SARS-CoV-2 vector vaccine, which eventually will trigger anti-vector immunity in the vaccinated individuals.
In this context Bates and al. characterize Ad49 (a species D adenovirus) in this study as a potential alternative adenoviral vector platform with low seroprevalence in the population. The authors compare luciferase expressing first generation vector based on Ad49 with Ad5 using transduction of several cell lines and in vivo biodistribution in mouse models following IV and intra-cranial injection. Furthermore, they perform assays to determine the mode of cell entry into different cell lines using a range of competition assays.
Overall Bates and al, provide some interesting observations for the biology of Ad49. However, several findings remain superficial and inconclusive leading to vague and speculative conclusions that could be validated by a more rigorous experimental design. Quantitative data are not always provided. Furthermore, the data presentation should be improved providing missing information and clarifying discrepancies between data presentation and description.
I have the following comments:
- Please check figure numbers (e.g. Figure 4 is before Figure 3; line 187 Figure 1B instead of 1C) and revise typos in the text (e.g. lines 92, 129, 230, 236, 268, 295…).
- In general, please avoid ambiguous language (e.g. line 190, 217, 246, 288, 306).
- It may be helpful to introduce the role of “FX” in liver tropism in the introduction.
- For the transduction experiments the authors use two different vector doses but never discuss the rational for this approach in the manuscript despite showing that the dose impacts on the transduction efficiency (e.g. for A549 cells). Please explain these differences.
- Why did the authors choose 10µg/ml as FX concentration and how many particles have been used in the experiment shown in Fig. 1C? Would a lower vector dose not provide stronger data on the FX effect in cell transduction?
- The “FX” effect on Ad49 infectivity for MDA 435 cells is mentioned but not discussed (and neither the absence of effect in the other cell lines). What distinguishes those cells from the other cell lines?
- Please add the Ad49 label to panels 2B and 2C
- Please explain “high” in the panel 2D, add an axis label and explain what 10^11 vp means?
- The analysis of cytokines in Fig. 2E is not convincing. I could not find any description of the methodology and the data presentation is unclear. The provided numbers have no units and it is not clear what experimental approach has resulted in these numbers. Furthermore, the provided stats indicate a large value variation (e.g. 56.12 fold increase with MIG, and 16.64 fold increase with IFN-g have a lower p-value than 2.49 fold with MCP1 or 1.88 fold with IP-10). Please explain in detail how these data have been obtained and chose an accurate graph presentation with proper error bars that permits data evaluation.
- Line 266 to 269: the in vivo bioluminescence assay is described as having “detectable levels” for Ad49 but “lower than with HAdV-C5”. Please add a proper quantification (for all bioluminescence data).
- Figure 5A: please provide proper color code for the right panel and add a Comassie gel to assess the purity/quality of the respective FK protein preps. To conclude that the AD49FK interferes with CAR binding (line 285) the authors should include a control with a known CAR-independent vector (perhaps AdB35 used in Fig. 5E?). The assumption would be that the Ad49 FK does not block such a vector.
- Figure 5B: please explain “ve”. Why are three wells per virus shown?
- Figures 5C and 5D: The authors write that neither neuraminidase nor heparinise treatment had a significant effect on transduction for Ad49. The p-value in Fig. 5C indicates significance. Please explain.
- Figure 5E: authors write that “HAdV-49 infection was reduced significantly in all three cell lines” (line 303) which is not the case in the figure stating no significance (n.s.) for A549 transduction. Please double check your statistical analysis and/or the text. In general, the experiments provided in Fig. 5 do not allow to deduce the receptor usage for Ad49 nor do they exclude a specific receptor convincingly. To support their claim of a brought receptor usage for Ad49 the authors may consider combinations of inhibitions with appropriate control vectors to gradually reduce Ad49 infectivity vs. specific inhibition of the controls. Alternative experimental approaches with a clearer readout should be considered.
Author Response
Author Comment: We thank the reviewer for their detailed and thorough review of our manuscript. We have taken their comments on board and this has resulted in a much improved version of the manuscript. We are grateful for these comments, which we respond to in detail, below. Please note that the line numbers stated refer to the non-tracked (“clean”) version of the document.
Adenovirus (Ads) have a well-defined and low pathogenic profile paired with technological accessibility. This makes adenoviruses powerful therapeutic tools for vector-based strategies. In contrast adenoviruses infections are widespread and a large part of the population has pre-existing immunity (antibodies) against the most common serotypes, notably species C adenoviruses. This can conflict with adenovirus vector applications, a problematic that will be further increased through the use of adenovirus as anti SARS-CoV-2 vector vaccine, which eventually will trigger anti-vector immunity in the vaccinated individuals.
In this context Bates and al. characterize Ad49 (a species D adenovirus) in this study as a potential alternative adenoviral vector platform with low seroprevalence in the population. The authors compare luciferase expressing first generation vector based on Ad49 with Ad5 using transduction of several cell lines and in vivo biodistribution in mouse models following IV and intra-cranial injection. Furthermore, they perform assays to determine the mode of cell entry into different cell lines using a range of competition assays.
Overall Bates and al, provide some interesting observations for the biology of Ad49.
Author comment: We thank the review for their synopsis of our manuscript and describing the observations in the manuscript as interesting.
However, several findings remain superficial and inconclusive leading to vague and speculative conclusions that could be validated by a more rigorous experimental design.
Author comment: In line with this comment and those which follow, we have modified the language to be less vague and speculative. Given we have needed to provide responses within a narrow time frame, we have not been able to perform additional studies. We have, however, added additional quantification as suggested, which has allowed us to be less speculative in our language.
Quantitative data are not always provided. Furthermore, the data presentation should be improved providing missing information and clarifying discrepancies between data presentation and description.
Author comment: we agree with the reviewer. In line with their comment, we have added quantification as requested, and included additional methods. In addition, we have added individual plots for our Luminex analysis, as requested, as supplemental files.
I have the following comments:
- Please check figure numbers (e.g. Figure 4 is before Figure 3; line 187 Figure 1B instead of 1C) and revise typos in the text (e.g. lines 92, 129, 230, 236, 268, 295…).
Author comment: We apologise for the misnumbering of the figures. We chose to change the figure order and failed to update the numbering. This has been amended. Similarly, typos in the text have been corrected.
- In general, please avoid ambiguous language (e.g. line 190, 217, 246, 288, 306).
Author comment: We agree and have updated the language as suggested to be less ambiguous.
- It may be helpful to introduce the role of “FX” in liver tropism in the introduction.
Author comment: We agree that this would be helpful, and now include a section in the introduction to introduce the role of FX in liver transduction (lines 55-66).
- For the transduction experiments the authors use two different vector doses but never discuss the rational for this approach in the manuscript despite showing that the dose impacts on the transduction efficiency (e.g. for A549 cells). Please explain these differences.
Author comment: The doses used are relatively standard “low” (1,000 vp/cell) and “high” (10,000 vp/cell). We routinely use such doses as standard in our transduction assays. We have clarified this in the revised text (line 209 – 210).
- Why did the authors choose 10µg/ml as FX concentration and how many particles have been used in the experiment shown in Fig. 1C? Would a lower vector dose not provide stronger data on the FX effect in cell transduction?
Author comment: We always use FX at a concentration of 10μg/ml as this represents the physiological concentration of FX in the blood and is therefore the most physiologically relevant dose to use. We also know from our extensive previous research that this dose works well. We clarified this in the revised manuscript (line 221). We also clarified the dose used in figure 1C (5000 vp/cell, line 233).
- The “FX” effect on Ad49 infectivity for MDA 435 cells is mentioned but not discussed (and neither the absence of effect in the other cell lines). What distinguishes those cells from the other cell lines?
Author comment: It is noteworthy that in addition to the enhanced transduction by Ad49 in the presence of FX, the FX effect in MDA 435 cells is also higher for Ad5 than in any other cell line tested. This may reflect higher levels of HSPGs (the cellular receptor for the Ad: FX complex) on this cell line compared to other cell lines and may also provide an explanation for why we observe a FX effect with Ad49 in this cell line also compared to other cell lines. However, since we have not validated this as a reason, we have simply noted this this may reflect an altered/increased expression of HSPGs on this cell line (line 223-5)
- Please add the Ad49 label to panels 2B and 2C
Author comment: This has been done.
- Please explain “high” in the panel 2D, add an axis label and explain what 10^11 vp means?
Author comment: We apologise, we have clarified that the only dose shown is 1011 viral particles per mouse. The work “high” was erroneous and has been removed. An axis label has also been added (“organ”).
- The analysis of cytokines in Fig. 2E is not convincing. I could not find any description of the methodology and the data presentation is unclear. The provided numbers have no units and it is not clear what experimental approach has resulted in these numbers. Furthermore, the provided stats indicate a large value variation (e.g. 56.12 fold increase with MIG, and 16.64 fold increase with IFN-g have a lower p-value than 2.49 fold with MCP1 or 1.88 fold with IP-10). Due to wide variation. Please explain in detail how these data have been obtained and chose an accurate graph presentation with proper error bars that permits data evaluation.
Author comment: We apologise for the lack of method. This has been added in the revised manuscript (lines 177-192). We have included units which were missing (pg/ml) and details of the experimental approach (a standard Luminex based cytokine array). Since the outputs are numerous, showing all the data as graphs becomes unwieldy and a very large figure, hence why we attempted to summarise this data as a table. We agree with the reviewer that there is value in being able to access the individual bar charts to assess whether the fold changes related to low baseline levels of cytokines, and therefore provide an extensive series of bar charts (with error bars) as a supplemental file showing the data for each cytokine assessed individually. We believe this provides the reader with the optimal opportunity to appropriately assess the changes in individual cytokines, without making the main manuscript unwieldy. We hope that this is acceptable to the reviewer.
- Line 266 to 269: the in vivo bioluminescence assay is described as having “detectable levels” for Ad49 but “lower than with HAdV-C5”. Please add a proper quantification (for all bioluminescence data).
Author comment: We include full quantification for all IVIS images shown (given beneath each of the respective images). This was a helpful suggestion, and we thank the reviewer for this.
- Figure 5A: please provide proper color code for the right panel and add a Comassie gel to assess the purity/quality of the respective FK protein preps. To conclude that the AD49FK interferes with CAR binding (line 285) the authors should include a control with a known CAR-independent vector (perhaps AdB35 used in Fig. 5E?). The assumption would be that the Ad49 FK does not block such a vector.
Author comment: We added the appropriate colour code to the right panel, and again thank the reviewer for their diligence in noting this omission. The suggestion to add a Coomassie panel to show that the recombinant fiber knobs are the right size and trimerize appropriately is an excellent suggestion, and we have added this data (revised figure 5A) to show that the proteins produced are the correct size as a trimer (~60-70 kDa) and when heated, as a monomer (~20 kDa). Whilst we agree that an additional experiment looking at whether Ad49 knob protein does not impede transduction of viruses using CD46 (such as Ad35), we are unable to perform this experiment due to time restrictions and the fact that these vectors were provided under historic MTA to Glasgow University (from Crucell, now Janssen) which has now lapsed. Janssen contact (Jerome Custers) approved the submission of this manuscript (but does not require to be listed as a co-author).
- Figure 5B: please explain “ve”. Why are three wells per virus shown?
Author comment: +ve and -ve depict positive and negative control treatments for red blood cell lysis. Negative control is PBS (no lysis), positive control is PBS containing 0.2% Tween-20 (PBST) to lyse red blood cells. Three wells are shown simply to visualise the n=3 technical replicates. We updated the figure legend to reflect this (lines 346-7).
- Figures 5C and 5D: The authors write that neither neuraminidase nor heparinise treatment had a significant effect on transduction for Ad49. The p-value in Fig. 5C indicates significance. Please explain.
Author comment: The reviewer is correct. We have modified the text accordingly (lines 329-332).
- Figure 5E: authors write that “HAdV-49 infection was reduced significantly in all three cell lines” (line 303) which is not the case in the figure stating no significance (n.s.) for A549 transduction. Please double check your statistical analysis and/or the text. In general, the experiments provided in Fig. 5 do not allow to deduce the receptor usage for Ad49 nor do they exclude a specific receptor convincingly. To support their claim of a brought receptor usage for Ad49 the authors may consider combinations of inhibitions with appropriate control vectors to gradually reduce Ad49 infectivity vs. specific inhibition of the controls. Alternative experimental approaches with a clearer readout should be considered.
Author comment: The reviewer is correct that further work is required to firmly define whether Ad49 uses a combination of known Ad receptors and/or uses a novel, undocumented receptor. Unfortunately, time and MTA restrictions (as mentioned above) limit our ability to perform these additional studies. Instead, we mention suggest these studies should be performed as part of future studies to fully define the (rather complex) receptor usage of Ad49. (lines 391 – 394)
Reviewer 2 Report
This paper contains an excellent characterization of human adenovirus type 49 as 2 a vector for therapeutic applications. Minor comments: 1. In vitro transduction correlates in general very, very poorly with in vivo transduction. This is again confirmed in this study. 2. The data in Figure 2E indicate that innate immune responses were more pronounced following Ad49 compared to an equivalent dose of Ad5. Is this related to spleen tropism? 3. Have the authors any idea which cells are transduced in the lung and spleen?Author Response
This paper contains an excellent characterization of human adenovirus type 49 as a vector for therapeutic applications.
Author comment: We thank the reviewer for their positive assessment of our manuscript.
Minor comments:
- In vitrotransduction correlates in general very, very poorly with in vivo transduction. This is again confirmed in this study.
Author comment: We absolutely agree with the reviewer and make this point clearly in the revised paper.
- The data in Figure 2E indicate that innate immune responses were more pronounced following Ad49 compared to an equivalent dose of Ad5. Is this related to spleen tropism?
Author comment: We suspect the reviewer is correct and this relates to the increased uptake of virus by immune cells within the spleen. We mention this in the manuscript, but since we have not fully characterised the cell types which are taking up virus, we have been careful not to over-interpret these findings (lines 403-407)
- Have the authors any idea which cells are transduced in the lung and spleen?
Author comment: Unfortunately, IHC was never performed on these samples, but we agree with the reviewer that this will be important to dissect in order to fully characterise the increased innate immune responses to Ad49. We mention this as future work in the revised manuscript (lines 403-407).
Round 2
Reviewer 1 Report
I thank the authors for taking the time to address my comments. I understand the restrictions on further experimentation for the present study and appreciate the revisions and clarifications in data presentation, which the authors provide. The manuscript is much clearer and does for an interesting (in its purest sense) read.